# The Influence of Low-Temperature Food Waste Biochars on Anaerobic Digestion of Food Waste

**DOI:** 10.3390/ma15030945

**Published:** 2022-01-26

**Authors:** Kacper Świechowski, Bartosz Matyjewicz, Paweł Telega, Andrzej Białowiec

**Affiliations:** Department of Applied Bioeconomy, Wrocław University of Environmental and Life Sciences, 37a Chełmońskiego Str., 51-630 Wrocław, Poland; 110829@student.upwr.edu.pl (B.M.); pawel.telega@upwr.edu.pl (P.T.); andrzej.bialowiec@upwr.edu.pl (A.B.)

**Keywords:** methane fermentation, biogas, biomethane, biochar, pyrolysis, hydrothermal carbonization, biochemical methane potential, biogas production kinetics

## Abstract

The proof-of-the-concept of application of low-temperature food waste biochars for the anaerobic digestion (AD) of food waste (the same substrate) was tested. The concept assumes that residual heat from biogas utilization may be reused for biochar production. Four low-temperature biochars produced under two pyrolytic temperatures 300 °C and 400 °C and under atmospheric and 15 bars pressure with 60 min retention time were used. Additionally, the biochar produced during hydrothermal carbonization (HTC) was tested. The work studied the effect of a low biochar dose (0.05 g_BC_ × g_TSsubstrate_^−1^, or 0.65 g_BC_ × L^−1^) on AD batch reactors’ performance. The biochemical methane potential test took 21 days, and the process kinetics using the first-order model were determined. The results showed that biochars obtained under 400 °C with atmospheric pressure and under HTC conditions improve methane yield by 3.6%. It has been revealed that thermochemical pressure influences the electrical conductivity of biochars. The biomethane was produced with a rate (k) of 0.24 d^−1^, and the most effective biochars increased the biodegradability of food waste (FW) to 81% compared to variants without biochars (75%).

## 1. Introduction

### 1.1. Background

The implementation of a circular economy induces the new approaches of closing the loops of material and energy flows within the systems, including the new solutions for food waste management. The high biodegradability and high biogas potential of food waste may be utilized for both biogas and organic fertilizer production. The biogas yield may be enhanced, and the fertilizer quality may be improved by the addition of biochars derived from food waste. It may bring added value to food waste, a component of municipal solid waste (MSW) sustainable management. Progressing economic development is conducive to an increase in waste production. MSW causes environmental problems such as water, air, and soil pollution, loss of biodiversity, and resource depletion, and over-use of land [1]. To counteract the negative waste effects and to counteract resource depletion, the European Union (EU) goes to a circular economy, where waste becomes a new resource. According to the directive 2008/98/EC on waste [2], EU states should move towards a circular economy by achieving targets for preparing, reusing, and recycling MSW. These targets were set to a minimum of 55%, 60%, and 65% (by weight) by 2025, 2030, and 2035, respectively [2]. To meet the directive targets and goals of the circular economy, the Council of the European Union adopted a rule for the collection of bio-waste. By 2023, all EU states must collect bio-waste separately or recycle it at the source (home composting) [3].

The bio-waste term refers to biodegradable garden and park waste, food and kitchen waste from households, restaurants, caterers, and retailers, and comparable waste from food processing plants [2]. Bio-waste accounts for about 30% of the MSW stream and about 60% of bio-waste is made from food waste (FW) [4].

### 1.2. Bio-Waste Processing Methods

Currently, in the EU, MSW (containing bio-waste from households) are mainly processed in the mechanical-biological treatment plants (MBT). In the MBT, in the first step, waste is treated mechanically by screening to separate fractions’ streams. An undersize fraction constitutes mainly minerals and wet organic waste, while an oversize fraction consists of plastics and other flammable materials. The screening process is not perfect, and, therefore, part of plastics and other flammable materials go to the undersize fraction, while some organic waste stays in the oversize fraction. As a result, an undersize fraction is unfit for organic recycling, and plastics quality is lower in comparison to plastics collected separately at the source. After screening, the undersize fraction is processed by composting or anaerobic digestion to stabilize, where waste is converted into a low-grade compost-like output (CLO), which must be landfilled [5,6].

On the other hand, when MSW are collected separately, the recycling rate of materials increases, and organics recycling of bio-waste is possible. Waste streams collected separately have higher quality than mixed [5,6], and bio-waste can be converted by composting or anaerobic digestion to fertilizer. In both processes, microorganisms are used to break down organic matter. Compositing is the process under controlled conditions in the presence of oxygen, at an appropriate temperature and humidity of ~60%. Depending on composting technology, it may be done in pits, by piling and heaping [7], or in closed reactors with forced aeration also known as in-vessel systems [8]. During composting, organic matter can heat itself to 70 °C at the thermophilic phase, ensuring the destruction of pathogens [7]. The process also leads to a decrease in the mass and volume because of water evaporation and organic matter decomposition. Besides composting advantages like low-cost technology and easy process control, several drawbacks exist. The process requires external energy for heaps turning and/or aeration, and when out of a vessel system technology is used, gaseous and liquid emissions occur. Processing of green waste results in CO, CO_2_, CH_4_, H_2_, NH_3_, N_2_O, CH_4_, and volatile organic compounds (VOCs) emissions which cannot be avoided [9]. Therefore, if composting does not follow in closed reactors, a better option for biowaste processing is anaerobic digestion (AD). 

Methane fermentation is a decomposition of organic matter under an oxygen-free atmosphere by anaerobic microorganisms at 37 °C or 55 °C. The main process product is flammable biogas consisting of CO_2_ and CH_4_ about 1:1, and digestate residues that can be used as fertilizer or solid fuel as well. Similarly, to composting, a lot of different technologies exist. For an organic fraction of MSW, (i) solid-state anaerobic digestion, (ii) continuous digestion with thermophilic conditions, and (iii) plug flow and continuous stirring tank rectors [6] are the most suitable. Though investment costs are much higher for anaerobic processes compared to aerobic ones, surplus energy production, comparable quality fertilizer, and almost zero emissions are plays in favor of AD [10]. Therefore, biogas plants will gradually replace composting ones. 

### 1.3. Problems with AD of Bio-Waste 

Due to a variable of bio-waste composition, conducting the AD process entails certain difficulties. To maintain biogas production at a stable level, many monitoring parameters need to be taken into count (feedstock size, total solids, volatile solids, pH value, ammonium nitrogen, volatile fatty acids (VFA), redox potential, alkalinity ratio, biogas composition (CH_4_, CO_2_, H_2_ and H_2_S), temperature, trace elements concentration, organic loading rate (OLR), and hydraulic retention time (HRT)). As a result, trained workers with laboratory equipment are needed [11]. Lack of concise process control and optimization of bio-waste composition lead to harmful intermediate compounds’ production and process instability. It is due to organic waste nature. Most FW has acidic pH which consumes digested feedstock alkalinity and is quickly decomposed during the hydrolysis phase. Quick decomposition with a combination of high protein and lipids content leads to rapid generation and accumulation of ammonia (NH_3_), and VFAs over inhibitory levels [12]. Though high VFA concentration does not have to inhibit the process since VFAs are essential nutrients for bacteria growth, pH value needs to be kept at an optimal level to balance the inhibitory effects of VFAs and NH_3_ [13]. As a result of difficulties, AD of bio-waste (especially FW) is often performed at a low OLR of 2–3 g_COD_ × (L × d)^−1^ [12]. For that reason, different substances improving process stability and performance are added [13]. One such substance getting attention recently is biochar. 

Biochar is considered as the material improving the methane fermentation process [14]. Biochar can absorb compounds such as H_2_S and CO_2_, and it also has the potential to mitigate the inhibition of ammonia and acids. It also creates an optimal environment for the growth of microorganisms, which results in faster colony development and higher biogas yield. The effect of biochar addition (positive or negative) depends on the specific situation like reactor type (batch, continuous) substrate type, type of fermentation, type of the biochar, and others [14]. 

The biochar is produced from organic materials during thermal processing at temperatures above 300 °C in a free oxygen atmosphere. Depending on conditions, the process is called torrefaction (200–320 °C), pyrolysis (>300 °C) [15], or hydrothermal carbonization (180–320 °C) [16]. Besides temperature, other parameters specify these processes, inter alia residence time, pressure, and initial moisture. Torrefaction and pyrolysis are performed at atmospheric pressure for pre-dried materials, while hydrothermal carbonization is performed at overpressure for wet materials. Each process has pros and cons and is used for different materials and purposes. The amount and quality (desired properties) of carbonaceous material obtained from thermal processing depends on feedstock type and process conditions. In general, the higher the process temperature, the more energy-consuming the thermal processing, and the lower amount of biochar is produced in favor of the yield of other products (liquid and gases) [15,16,17]. Therefore, low-temperature biochars produced with lower energy demand than under high-temperature pyrolysis may be considered as a sustainable source of structural additive for FW AD. The scientific question on its influence on AD performance may be derived.

### 1.4. Study Aim

All the advantages of the AD process improvement by biochar addition have not been fully explored because biochar can be produced from various substrates, under different conditions, and various substrates can be processed by AD. Additionally, the application of biochar produced from the same materials as being processed under AD has been rarely studied [18]. In this work, five low-temperature biochars that potentially could be made using residual heat from biogas combined heat and power units (300–400 °C) were produced and used to enhance the AD of FW. Moreover, biochars were produced from the substrate (here food waste) under torrefaction, low-temperature pyrolysis, and hydrothermal carbonization conditions. 

## 2. Materials and Methods

### 2.1. Materials

#### 2.1.1. Inoculum Preparation 

As inoculum for biochemical methane potential tests, digestate from the 1 MW_el_ commercial agricultural biogas plant (Bio-Wat Sp. z o.o., Świdnica, Poland) was used. The biogas plant is operating on wet (dry mas < 10%) and mesophilic conditions (37 °C). The digestate was collected to plastic canisters and was taken to the laboratory where it was stored at room temperature for ~24 h. The next day, the digestate was filtered through gauze to separate liquid from solid particles: unprocessed substrate, plastics, etc. Then, the liquid digestate was stored in the climate chamber (Pollab, model 140/40, Wilkowice, Poland) at 4 °C before the biochemical methane potential test. 

#### 2.1.2. Food Waste Preparation

The food waste mixture for biochemical methane potential tests was prepared from food purchased in the grocery store. The mixture consists of 3.67% of orange, 8.67% of banana, 7.33% of apple, 1.33% of lemon, 24.33% of potatoes, 4.67% of onion, 3.33% of salad, 3.33% of cabbage, 2.33% of tomatoes, 6% of rice, 6% of pasta, 3% of bread, 3% of meat, 12% of fish meat, and 11% of cheese by fresh mass. The fresh food waste mixture had 64.2% of moisture content (MC), while volatile solids (VS) constituted 95.8% of dry mass. The ash content (AC) of the mixture was 4.2%. The FW composition was based on the work of Valta et al. [19]. The properties of moisture content, total solids (TS), volatile solids (organic matter content), and ash content, of used food materials, and mixture composition per fresh, dry, and volatile solids percentage share bases are presented in Table 1. 

FW components were dried in the laboratory dryer (WAMED, model KBC-65W, Warsaw, Poland) at 105 °C and shredded. Drying time differed depending on the food type. Then, dry food was ground through a 1 mm screen using a laboratory knife mill (Testchem, model LMN-100, Pszów, Poland). Ground FW samples were stored in plastic string bags, at room temperature. The mixture for AD was prepared from ground dry food materials according to data presented in Table 1. To ensure mixture homogeneity, one portion of 1 kg was prepared before the biochemical methane potential test. In addition, all tests were done using this mixture. 

#### 2.1.3. Low-Temperature Biochar Preparation and Analyses

The low-temperature biochars, low-temperature and low-pressure biochars, and low-pressure hydro-char were produced using a prototype batch laboratory reactor (WUELS, RBMT2020-1.1, Wrocław, Poland) presented in Figure 1. A full reactor design description is available elsewhere [20]. In short, the reactor is steel-made, an air-tight vessel of 22.3 dm^3^, wrapped in a 3 kW heating jacket and insulations (4). The process gas can be released by the upper (6) or lower valve (8). In this study, gas was released by the upper valve and went through a cooler that kept its temperature below 200 °C (to protect the manometer) (1). 

The biochars were produced from a dry FW mixture at 300 °C and 400 °C in 60 min, at atmospheric pressure, and overpressure of 15 bars. For each process, the residence time of 60 min was counted since the setpoint temperature inside the reactor was reached. For the process at overpressure, when the pressure in the reactor increased over 15 bars, it was released manually up to 14 bars. An exemplary biochar production parameters’ diagram is presented in Figure A1. The outer reactor wall temperature was around 150 °C higher than the setpoint temperature (inside the reactor). For low-pressure hydrothermal carbonization (15 bars), a dry FW mixture was mixed with water to obtain 64.2% moisture content (to simulate the initial moisture of FW). The setpoint temperature for hydrothermal carbonization was 280 °C.

For each process, a total sample mass of 250 g was used. Each sample was divided into five smaller samples of ~50 g that were placed into aluminum trays that next were covered with aluminum foil. Then, the five trays were placed evenly inside the reactor. The reason for sample dividing was to place it in a different part of the reactor to assure better heat transfer from the reactor’s walls to samples. The reason for covering trays with aluminum foil was to avoid sample incineration at the initial stage where some air could have been present in the reactor.

After 60 min, since the setpoint temperature inside the reactor was reached, the heating jacket was turned off. Additionally, in the case of overpressure processes, the upper valve has been opened to release pressure. Then, the reactor was left to cool down. After cooling down to room temperature, samples were removed. The difference between the initial and end mass of solids was used to calculate the mass yield of the biochar production following Equation (1):(1)$$MY=\frac{m_{b}}{m_{r}}\;\times \;100$$
where:

MY—mass yield, %;$m_{b}$—dry mass of biochar after the process, g,$m_{r}$—dry mass of material before process, g.

Produced biochars were analyzed for specific surface area (BET), total pore volume <50 nm (Vt), and average pore size <50 nm (L) by adsorption analyzer (Micromeritics, ASAP 2020, Norcross, GA, USA).

### 2.2. Methods

#### 2.2.1. Biochemical Methane Potential Test

Biochemical methane potential (BMP) tests were performed using an automatic methane potential test system (BPC Instruments AB, AMPTS^®^ II, Lund, Sweden) presented in Figure 2. The system consists of 15 reactors (500 mL) with agitation (2) placed in water batch (1), gas volume meters (4) as well as a built-in data acquisition system that can be displayed on PC (5). Due to the presence of CO_2_ absorption units filled with NaOH solution (4), only CH_4_ volume was measured.

A biomethane potential test took 21 days and was performed twice. Each replication consists of two reactors filled with digestate; two reactors filled with digestate and FW, and two reactors filled with digestate, FW, and biochar according to the matrix presented in Table 2. For each reactor, 300 g of liquid digestate was used. For each reactor (excluding the first two), 3.96 g of dry FW mixture was added, and for reactors with BC, 0.1982 g of dry biochar was added. As a result, the substrate to inoculum ratio (SIR) was 0.4 by VS (or 0.25 by TS), the total solids in the reactors were 6.53–6.59%, and biochar share in FW was 5% (by total solids). At the beginning and end of the test, ph and electrical conductivity (EC) was measured using a ph/EC meter (Elmetron, CPC-411, Zabrze, Poland).

The SIR of 0.4 was chosen due to works of [21,22], which show that, for FW, the optimal SIR varies from 0.33 to 0.5, while a 5% BC share in food waste by TS was chosen due to our previous work [18]. In addition, a 5% share of biochar addition considered in the current study is equal to biochar addition of 0.05 g_BC_ × g_TSsubstrate_^−1^, or 0.65 g_BC_ × L^−1^.

#### 2.2.2. Materials and Process Residue Analysis

All material used in the study was subjected to moisture content, total solids, volatile solids, and ash content determination [23]. The moisture content and total solids were determined using the laboratory dryer (WAMED, model KBC-65W, Warsaw, Poland), according to the PN-EN 14346:2011 standard [24], while volatile solids and ash content were determined using the muffle furnace (SNOL, 8.1/1100, Utena, Lithuania) according to the PN-EN 15169:2011 standard [25]. Additionally, biochars were analyzed for pH and EC. The measurements were performed in measured in solution: 1 g of dry mass to 10 mL of deionized water, after 30 min since being mixed [26].

FW mixture was additionally subjected to ultimate analysis for determination of the elemental composition (C, H, N, S, O). The ultimate analysis was performed using a CHNS analyzer (PerkinElmer, 2400 CHNS/O Series II, Waltham, MA, USA) according to 12902:2007 [27]. The oxygen content was calculated by the difference according to Equation (2):(2)O=100−C−H−N−S−AC
where:

O—oxygen % share in dry mass, %;C—carbon % share in dry mass, %;H—hydrohen % share in dry mass, %;S—sulfur % share in dry mass, %;AC—ash % share in dry mass, %.

The elemental composition was used for the calculation of theoretical biogas composition and the theoretical biochemical methane potential (TBMP). Calculations were done according to Boyle modification of Buswell and Mueller stoichiometric formulas, Equation (3) [28]:
(3)$$\begin{array}{ll} C_{a}H_{b}O_{c}N_{d}S_{e}+ & \left(a-\frac{b}{4}-\frac{c}{2}+\frac{3d}{4}+\frac{e}{2}\right)H_{2}O\\ & \to \left(\frac{a}{2}+\frac{b}{8}-\frac{c}{4}-\frac{3d}{8}-\frac{e}{4}\right)CH_{4}+\left(\frac{a}{2}-\frac{b}{8}+\frac{c}{4}+\frac{3d}{8}+\frac{e}{4}\right)CO_{2}+dNH_{3}+eH_{2}S \end{array}$$
where:$C_{a}H_{b}O_{c}N_{d}S_{e}$—elemental composition of the substrate, C—carbon, H—hydrogen, O—oxygen, N—nitrogen, S—sulphury, and *a*, *b*, *c*, *d*, *e* stands for molar % share of specific elements of the volatile solids of biomass [29].$H_{2}O$—water needed for substrate decomposition, mol;$CH_{4}$—methane, mol;$CO_{2}$—carbon dioxide, mol;$NH_{3}$—ammonia, mol;$H_{2}S$—hydrogen sulfide, mol.

The mols of biogas products ($CH_{4}$, $CO_{2}$, $NH_{3}$,$\;H_{2}S$) were recalculated for volume in standard conditions (*p* = 1013.25 hPa, *T* = 273.15 K) by multiplication obtained mols by 22.415 obeying Avogadro’s law. Knowing the elemental composition of substrates and the molar mass of each element, the mass of 1 mol of the substrate was calculated. Next, the volume of each gas component was divided by the mass of 1 mole of substrate used for its production, providing a result in dm^3^ per gram of dry substrate. Then, knowing the volatile solids of a substrate, results were recalculated to dm^3^ of gas per gram of volatile solids of a substrate.

Additionally, the FW biodegradability was calculated using data of cumulative methane production and theoretical maximum methane production following Equation (4) [30], and CH_4_ production effect, Equation (5):(4)$$BD=\frac{EBMP}{TBMP}\;\times \;100$$
where:

BD—biodegradability of FW obtained in the methane fermentation process, %;EBMP—experimental biochemical methane potential, ml × g_VS_^−1^;TBMP—theoretical biochemical methane potential, ml × g_VS_^−1^;

(5)$$CH_{4}production\;effect=\frac{CH4_{with\;BC}-CH4_{without\;BC}}{CH4_{without\;BC}}\;\times \;100$$where:

$CH_{4}production\;effect$—change of CH_4_ produced after biochar addition to the process, %;$CH4_{with\;BC}$—CH_4_ produced from a sample without biochar added, ml;$CH4_{without\;BC}$—CH_4_ produced from a sample with biochar added, ml.

#### 2.2.3. Methane Production Kinetics

The results of the BMP test were subjected to kinetics determination. The first-order equation (Equation (6)) was used to provide information about the rate of methane production and the estimated value of maximum methane production potential with the application of Statistica 13.0 software (TIBCO Software Inc., Palo Alto, CA, USA). Afterward, the methane production rate was calculated (Equation (7)) [18]:


(6)
$$BMP=EBMP_{e}\;\times \;\left(1-e^{\left(-k\;\times \;t\right)}\right)$$


(7)$$r=k\;\times \;EBMP_{e}$$where:

BMP—the cumulative methane production obtained from a substrate after time t, ml_CH4_ × g_VS_^−1^;$EBMP_{e}$—the estimated value of experimental maximum methane production obtains from a substrate, ml_CH4_ × g_VS_^−1^;k—constant reaction rate, d^−1^;t—process time, d;r—methane production rate, ml_CH4_ × (g_VS_ × d)^−1^.

#### 2.2.4. Statistical Analysis of Biochar Effect

To check if biochar addition had a statistically significant effect (positive or negative) on the methane fermentation, the one-way analysis of variance with post-hoc Tukey tests was performed at the level of α = 0.05, with the application of Statistica 13.0 software (TIBCO Software Inc., Palo Alto, CA, USA).

## 3. Results and Discussion

### 3.1. Substrate and Biochar Properties

The liquid digestate used for BMP had 7.86 of pH, 68.8 µS × cm^−1^ of EC, 94.7% of MC, 5.3% of TS, 59.3% of VS, and 40.7% of AC, while the FW mixture (substrate) used for BMP tests had 5.6% of MC, 94.4% of TS, 95.8 of VS, and 4.2% of AC (Table 1). The elemental analysis showed that FW mixture was characterized by 44–47.8%, 5.7–6.2%, 39.9–44.4%, 1.45–1.58%, 0.24–0.26% of C, H, O, N, S, respectively (by dry mass base). In addition, the FW mixture was characterized by a pH of 5.62 and EC of 3.6 mS × cm^−1^.

The five types of biochars were used depending on the production conditions as follows: temperature/time/pressure; however, the HTC280 means a hydrothermal carbonization process at 280 °C in 60 min. The biochars were characterized by MY ranging from 34.3% to 56.4% for 400/60/15 and HTC280, respectively (Table 3). The highest MY was noted in the case of HTC and the 300/60/15 process (Table 3). As result, for biochars with high MY, less substrate and energy are needed for their production in comparison to biochars with low MY. Nevertheless, in such a scenario, the substrate is less converted, and biochar may not have the desired properties [31]. Produced biochars had a relatively low volatile solid content compared to FW used for biochar production. On the other hand, biochars had a much higher ash content than the FW mixture. The ash content in biochar varied from 10.4% to 39.1%, while the FW mixture had only 4.2% of ash. The produced biochar was also analyzed for specific surfaces area (SSA) according to BET theory, total pore volume <50 nm (V_t_), and average pore size <50 nm (L). Moreover, produced biochars had a value of SSA ranging from 0.26 to 0.64 g × m^−2^, and pore size ranging from 5.2 to 7.1 nm (Table 3). The total pore volume ranged from 3.3 × 10^−4^ cm^3^ × g^−1^ to 8.2 × 10^−4^ cm^3^ × g^−1^, excluding 400/60/15 biochar that had V_t_ of 11.3 × 10^−4^ cm^3^ × g^−1^ (Table 3). The pyrolysis results in biochars’ pH increase from 5.62 to 8.61–10.75, except HTC280, for which pH decreased to 5.59. Except for biochar produced at 300 °C, all biochars had higher EC in comparison to the FW (Table 3).

The pore volume, pore size, specific surface area, pH, elemental composition, surface functional groups, electrical conductivity (EC), and cation exchange capacity (CEC) are considered as key biochar physicochemical properties which affect the AD and biogas production [32]. Porosity is considered a key factor to recognize the plausible relations with microbes in AD. The porosity is characterized in terms of the average diameter [33] and is described by three main pore type: micropores (<2 nm), mesopores (2–50 nm), and macropores (>50 nm). For activated carbon, a specific surface area of micropores may constitute up to 95% of the total SSA of activated carbon. As result, micropores decide about the adsorption capacity. On the other hand, mesopores significantly contribute to the adsorption of larger particles, such as dye or humic acids [34]. Generally, pores with a radius over 25 nm are considered transport pores, while pores smaller than 25 nm are considered adsorbing ones [35]. Besides absorption, pores provide a microorganism habitat for proliferating since the typical size of bacteria is 0.3 μm to 13 μm. The higher the SSA, the more effective biochar is in the interaction with the surrounding species [33]. The SSA of biochar varied significantly depending on substrate and process conditions. The SSA in activated carbons varies from 419 to 3102 m^2^ × g^−1^ [36], while for low-temperatures and not activated biochar (350–500 °C), it varies from 0.36 to 5.31 g × m^−2^. Moreover, pore volume and average pore size in such biochars vary from 10 × 10^−4^ to 80 10^−4^ cm^3^ × g^−1^, and 2.39 to 14.60 nm, respectively [37]. It means that biochars produced in the current study do not differ significantly in comparison with other biochars produced at similar temperatures but have incomparably smaller SSA in comparison to activated carbon.

Since electrically conductive materials (i.e., mineral particles, carbon materials) added to AD show a reduction in lag phase and increased methane production rates, electrically conductive materials found more attention. Conductive materials (i.e., biochar, graphite, activated carbon) added to AD can promote direct interspecies electron transfer (DIET) between syntrophic partners [38]. The DIET is an alternative to interspecies H_2_/formate transfer for syntrophic electron exchange between microbial species. In AD, some methanogens can receive electrons from other microorganisms by molecular electric connections or by conductive materials [39]. For that reason, materials with good electrical conductivity properties are assumed to help enhance methane fermentation. The biochar electrical conductivity can be measured in solid-state [40], as powder [41], or in water solution, like soil EC is measured [42]. The EC varies depending on the method, and therefore caution is needed when data are compared between studies. Nevertheless, results from the same method show that an increase in pyrolysis temperature increases EC value. In addition, this is due to higher carbonization and an increase in ash content [41]. Biochar EC values may vary from 0.04 mS × cm^–1^ to 54.2 mS × cm^–1^, and besides pyrolysis temperature, the feedstock affects EC as well [42]. These show that biochar produced in this study had relatively low EC (3.04–7.69 mS × cm^–1^) in comparison to biochars found in the literature.

The pH is an important factor affecting the BMP test results and will be described in more detail later. It is worth noting here that all biochars except HTC280 were alkaline, and their pH increased with process temperature, while HTC280 become more acidic. In addition, it is worth noting that pH did not change when pressure was applied, while EC increased, 3.04 vs. 3.57 mS × cm^–1^ for biochars made at 300 °C, and 4.53 vs. 7.69 mS × cm^–1^ for biochars made at 400 °C. This suggests that pressure may potentially be a parameter that can be used to modify EC. This finding should be further investigated.

### 3.2. Biochemical Methane Potential—Theoretical and Experimental

The effect of low-temperature biochar addition on the cumulative biomethane production process for 21 days was investigated (Figure 3). The result shows that the highest methane production was obtained for biochar from hydrothermal carbonization (HTC280) and biochar produced at 400/60/0. The control reactors obtained 347.9 ml_CH4_ × g_VS_^−1^, while reactors with biochars 400/60/0 and HTC280 had 360.1 ml_CH4_ × g_VS_^−1^ and 365.2 ml_CH4_ × g_VS_^−1^, respectively (Figure 3). The lowest value of BMP was obtained for reactors where biochar 400/60/15 was added (331.7 ml_CH4_ × g_VS_^−1^).

The theoretical biochemical methane potential of the food waste mixture was 460 ml_CH4_ × g_VS_^−1^ (Equation (3)). In addition, theoretical calculations showed that, for complete substrate conversion into biogas, 437 ml_CO2_ × g_VS_^−1^, 25 ml_NH3_ × g_VS_^−1^, and 2 ml_H2S_ × g_VS_^−1^ will be produced. The experimental BMP test for control samples after 21 days obtained 347.9 ml_CH4_ × g_VS_^−1^ (Figure 3) reaching 75.5% substrate biodegradation.

Experimental BMP values obtained in this study are lower than the BMP value for source-separated domestic FW collected in the EU, for which BMP ranges from 420 to 470 ml_CH4_ × g_VS_^−1^ [43]. Nevertheless, the theoretical potential is in this range, and most reactors reached BD over 75%, which suggests that BMP was done properly, especially since the processing time was only 21 days.

The CH_4_ production effect shows a difference between the value obtained from the control (D + FW) and the reactor with biochar (Table 4). When the value is greater than 0, biochar increased the methane production, while when the value is lower than 0, biochar decreased methane production in comparison to control. The biochar addition had a positive effect on methane production from FW. Only biochar 400/60/15 showed a decrease in methane production. For this biochar, all reactors produced less methane than control. For other biochars, mean value from the repetitions was generally positive, and more methane was produced than by control. Nevertheless, biochars produced at 300 °C led to a decrease in methane production in some repetitions. The highest methane production was obtained from reactors where 400/60/0 and HTC280 were added, 3.5%, and 3.6% respectively (Table 4). Among literature, various effects of biochar addition on methane production effect can be found. Results differ from total process inhibition to a several-fold increase in methane production. The effect is highly dependent on factors such as initial conditions of the batch test, used inoculum and substrate, the substrate to inoculum ratio, biochar dose, biochar type, and conditions of its production) [22,44,45,46,47]. Kaur et al. [47] added biochars produced at 550 °C and 700 °C from wood, oilseed rape, and wheat straw at a dose of 10 g_BC_⋅L^−1^ to co-fermentation of food waste and sewage sludge under a high SIR level of 11.5 by VS. As a result, cumulative methane production increased from 4.5% to 24%. In addition, the highest increase was observed for biochar made from wheat straw at 550 °C, and the lowest for oilseed rape produced at 700 °C [47]. On the other hand, Sunyoto et al. [22] added biochar made from pine sawdust at 650 °C to anaerobic digestion of food waste. Biochar doses of 8.3, 16.6, 25.1, and 33.3 gBC⋅L^−1^ were studied, and results showed that only a dose of 8.3 increased methane production by 6.2%, while others decreased methane production up to 12.9%. It is also worth noting that biochar doses that increased methane potential did not do it significantly, while biochar doses higher than 25.1 g_BC_⋅L^−1^ significantly decreased methane production (at the *p*-value of 0.002) [22]. Furthermore, the results of Zhang et al. [45] that conducted methane fermentation of FW at thermophilic conditions showed that the lowest of tested biochar doses (6 g_BC_⋅L^−1^) gave the highest cumulative methane production [45]. Because, in the current study, only one dosage of 0.65 g_BC_⋅L^−1^ was tested, and other research proved that a biochar dose of up to 10 g_BC_⋅L^−1^ can improve methane production, higher doses of 400/60/0 and HTC280 should be tested in the future.

The initial pH in all reactors with FW and biochar differed from 7.62 to 7.91, while EC differed from 56.1 to 67.9 µS × cm^−1^. After 21 days of the process, pH differed from 7.92 to 8.03, and EC differed from 68.7 to 77.7 µS × cm^−1^ (Table 4). For comparison, digestate alone had an initial pH and EC of 7.86, and 66.8 µS × cm^−1^, respectively, while, after 21 days, these parameters were 8 and 71.8 µS × cm^−1^, respectively (Table 4). The initial pH is an important parameter affecting methane yield in batch experiments, but no one value would show the correctness of the process [48]. The initial pH and then its changes during the process affect product yield, as optimal pH was reported value from 6.8 to 7.4 [49]. Anaerobic digestion is a four-stage process consisting of hydrolysis, acidogenesis, acetogenesis, and methanogenesis. The pH is crucial in each stage, and each of them required a different value. A positive correlation was found between the hydrolysis rate and pH [49]. The optimal pH for acidogenesis is 5.5–6.5 [50], while methanogenesis is effective when pH is around 6.5–8.2 (with optimum pH of 7.0) [51]. Even though methanogenesis is effective at 6.5, the methanogens’ growth rate is reduced significantly at a pH lower than 6.6 [52]. Therefore, the best result of AD can be obtained by a division process into two-stage hydrolysis with acidogenesis, and acetogenesis with methanogenesis [49]. The pH also affects the decomposition of total solids, and volatile solids in the reactor, as well as volatile fatty acid composition [53,54]. Nevertheless, in this study, biochar addition did not significantly change pH (*p* < 0.05), and as result, all reactors had similar conditions. Here it is worth noting that, for some reason, biochars with completely different pH, 10.19 vs. 5.59 for 400/60/0 and HTC280, respectively, showed the best methane production enhancement. The reason for that may be some other biochar properties that were not considered in this study. Maybe these biochars enhanced buffer capacity in the highest way despite different pH, and, as a result, provided better conditions for microorganism growth.

The EC shows the number of dissolved salts in solutions and is proportional to the quantity of these salts. The solutions with higher salt concentration have a greater ability to conduct an electrical current [42]. In the methane fermentation process, this parameter alone is rather useless. Nevertheless, EC can be used in online monitoring of biogas plants for prediction in advanced methane production of up to two days [55], or alkalinity [56]. As mentioned previously, conductive materials can enhance methane production by DIET. Nevertheless, in this study, biochar addition did not change the electrical conductivity of the solution significantly (*p* < 0.05); therefore, it is highly probable that DIET had no effect here.

Generally, biochar addition did not lead to significant (*p* < 0.05) changes in pH, and EC obtained biodegradability, substrate mass reduction, and amount of produced CH_4_. However, even though no statistically significant differences were found, results of biochar made at 400/60/0 and HTC280 showed to always have higher methane production than control, on average by 3.5% (Table 4). At first sight, it looks small; however, when the 1 MWe FW biogas plant working for 8000 h per year is considered, after the addition of BC, the additional 280 MWh of electricity may be produced. It is worth noting that usually biogas plants have problems with the utilization of heat, which in this case may be used for BC production.

### 3.3. Biomethane Production Kinetics

The mean kinetic parameters evaluated by the model for control (D + FW) were *k* = 0.240 d^−1^, $EBMP_{e}$ = 351.4 ml_CH4_ × g_VS_^−1^ and *r* = 84.43 ml_CH4_ × (g_VS_ × d)^−1^ (Table 5). All determined kinetics had a high determination coefficient (R > 0.99) (Table 5), which suggests that the used model fits the experimental data well. In general, the first-order model is used for quickly and abruptly stopping degradation substrates [57]. Furthermore, there was no need to use more sophisticated models like the modified Gompertz equation (good fitting when a lag phase is present), the mondo model (good fitting when gas production slowly declining at the end of the process), or two first-order equations (good fitting when two separate degradation profiles occur) [57] since here no such situation took place and biodegradation of over 75% was obtained in 21 days (Table 4).

The biochar addition changed the values of kinetic parameters slightly, but these changes were not statistically significant (*p* < 0.05). The highest constant production rate of biomethane was observed for 300/60/15 (*k* = 0.246 d^−1^), while the lowest for 400/60/15 (*k* = 0.229 d^−1^). Overall, 400/60/15 addition resulted in the worst kinetics, and the $EBMP_{e}$ and *r* were 334.22 ml_CH4_ × g_VS_^−1^ and 76.36 ml_CH4_ × (g_VS_ × d)^−1^, respectively. On the other hand, the best kinetics were obtained for 300/60/0 and 400/60/0 (Table 5). These results are a little confusing since the experiment showed that the highest methane production was for 400/60/0 and HTC280; nevertheless, this is probably due to a simplification of the model, which was not able to consider the increase in CH_4_ production after 17 days visible for HTC280 (Figure 3).

Overall, the results of methane production kinetics were determined accurately. The maximum methane potential and process kinetics are highly dependent on substrate, inoculum, equipment, and process conditions such as TS and pH. Deepanraj et al. [58] analyzed the kinetic of biogas production from kitchen waste at different TS concentrations (5–15%) and pH (5–9). The results of Deepanraj et al. [58] showed that first-order model kinetics (Gompertz model name by author) fit well to experimental data and had a determination coefficient >0.994. Moreover, results showed that the highest biogas production was obtained for TS = 7.5% and pH of 7 [58]. Is worth noting that these values are close to the ones used in this study (TS varied from 6.53 to 6.59%, and pH varied from 7.62 to 7.91). This suggests that those are important parameters for food waste anaerobic digesting and should be always considered when a BMP test of FW is prepared. There are pieces of evidence in the literature for which biochar addition can improve anaerobic digestion of food waste e.g., by improving process stability, decreasing lag phase, increasing methane yield, etc. Some theories described a process, how biochar enhances AD. Nevertheless, the abundance of food waste and used equipment/procedures lead to different AD enhancement results among studies—bearing in mind that biochar production consumes energy, and biochar transport to biogas plants costs as well. Different low-temperature biochars that potentially could be made using residual heat from biogas combined heat and power unit (CHP) (300–400 °C) were tested. It must be noted that biochars were made from a substrate used in a biogas plant and added to reactors at only one low dose (0.05 g_BC_ × g_TSsubstrate_^−1^, or 0.65 g_BC_ × L^−1^). The application of different BC doses might influence biomethane production more significantly. It should be further investigated.

## 4. Conclusions

Executed experiments, on the application of biochar produced from the same substrate as used for the anaerobic digestion (food waste) under different low-temperature and pressure conditions, indicated that:not all low-temperature biochars at the presented dose can improve biomethane production yield;the biomethane yield changes are visible for extreme cases. The worst biochar led to an average 4.5% CH_4_ decrease, while two of the best biochars increased CH_4_ production on average by 3.5%;biomethane production was improved on average by 3.5% by biochar made at 400 °C in 60 min at atmospheric pressure, and by low-pressure hydrochar produced at 280 °C, while the biodegradability of FW was higher than 81% in those variants;the theoretical CH_4_ potential of food waste was 460 ml_CH4_ × g_VS_^−1^, while the first-order constant reaction rate was *k* = 0.24 d^−1^;the FW thermal treatment pressure may influence the EC of biochar.

Further research is needed at low-temperature biochars since this study did not clearly reveal the dependence between low-temperature biochars addition and methane production yield. More trials with different biochar production pressure variants, biochar doses, and at different food waste concentrations should be performed for the validity of the low-temperature biochar application in AD.

## Figures and Tables

**Figure 1 materials-15-00945-f001:**
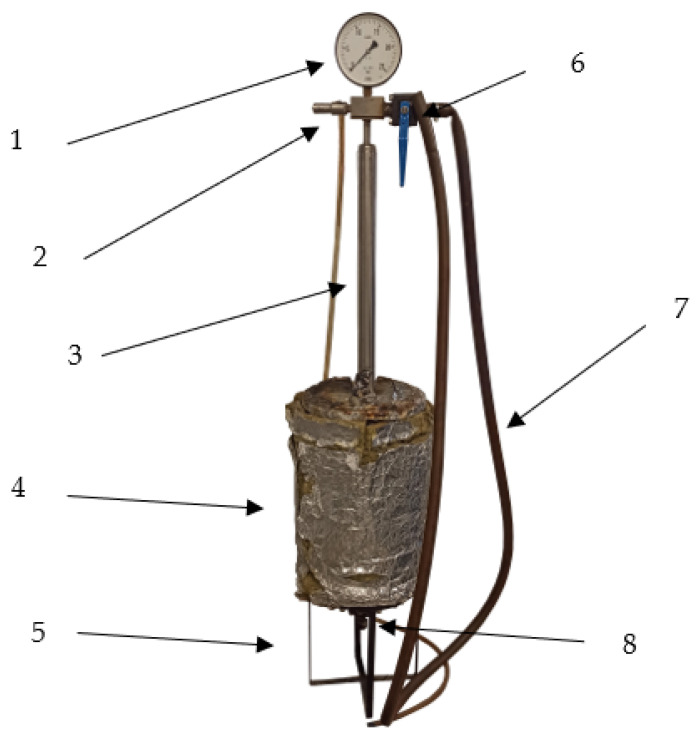
Reactor RBMT2020-1.1 used for biochar production, 1—manometer, 2—safety valve, 3—gas cooler, 4—reactor chamber wrapped by heating jacket and insulation, 5—stand, 6—upper valve, 7—exhaust gas pipe, 8—lower valve.

**Figure 2 materials-15-00945-f002:**
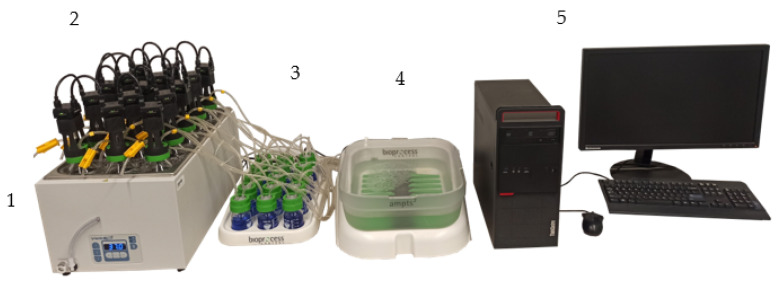
Biochemical methane potential test equipment AMPTS II, 1—water bath, 2—reactors with agitation, 3—CO_2_ absorption units, 4—gas volume meters, 5—computer.

**Figure 3 materials-15-00945-f003:**
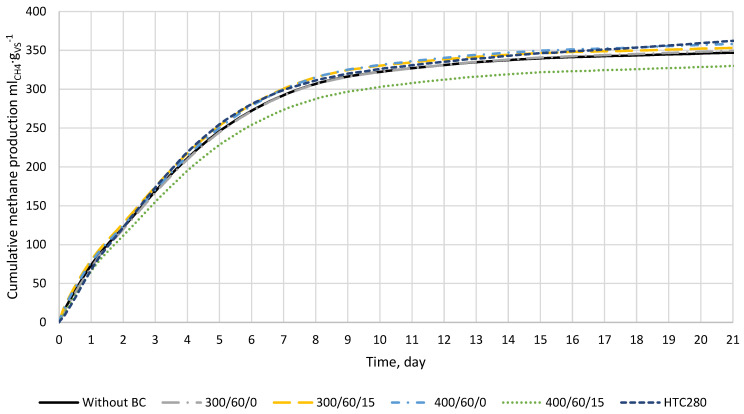
The biomethane production from food waste (*n* = 4). The results show CH_4_ production in ml per gram of food waste volatile solids, and the CH_4_ produced by inoculum (digestate) was subtracted.

**Table 1 materials-15-00945-t001:** Food waste properties and its share in food waste mixtures.

Material	Basic Properties				Share in Mixture		
	MC, % *	TS, % *	VS, % **	AC, % **	By Fresh Mass, %	by Dry Mass, %	by VS, %
Mixture	64.2	35.8	95.8	4.2	-	-	-
Orange	86.2	13.8	95.3	4.7	3.67	1.42	1.43
Banana	81.4	18.6	87.8	12.2	8.67	4.51	4.19
Apple	87.4	12.6	95.4	4.6	7.33	2.58	2.60
Lemon	85.4	14.6	93.5	6.5	1.33	0.55	0.54
Potatoes	61.6	38.4	93.1	6.9	24.33	26.11	25.73
Onion	89.2	10.8	93.4	6.6	4.67	1.41	1.40
Salad	94.9	5.1	85.7	14.3	3.33	0.48	0.43
Cabbage	92.2	7.8	91.6	8.4	3.33	0.72	0.70
Tomatoes	95.1	4.9	82.1	17.9	2.33	0.32	0.32
Rice	13.2	86.8	99.4	0.6	6.00	14.55	15.31
Pasta	11.6	88.4	95.5	4.5	6.00	14.84	15.00
Bread	22.5	77.5	95.2	4.8	3.00	6.50	6.54
Meat	69.8	30.2	96.0	4.0	3.00	2.53	2.57
Fish meat	81.7	18.3	95.5	4.5	12.00	6.12	6.19
Cheese	43.5	56.5	92.8	7.2	11.00	17.37	17.06

* as received base. ** as dry base.

**Table 2 materials-15-00945-t002:** Anaerobic digestion experiment matrix, D—digestate, FW—food waste, BC_—specific biochar derived under the following conditions: temperature, °C/residence time, min./pressure, bar.

Sample	Digestate	Food Waste Mixture	Biochar
D	+	-	-
D	+	-	-
D + FW	+	+	-
D + FW	+	+	-
D + FW + BC_300/60/0	+	+	+
D + FW + BC_300/60/0	+	+	+
D + FW + BC_300/60/15	+	+	+
D + FW + BC_300/60/15	+	+	+
D + FW + BC_400/60/0	+	+	+
D + FW + BC_400/60/0	+	+	+
D + FW + BC_400/60/15	+	+	+
D + FW + BC_400/60/15	+	+	+
D + FW + BC_ HTC280	+	+	+
D + FW + BC_ HTC280	+	+	+

D—digestate; FW—food waste mixture, BC_300/60/0—biochar produced at 300 °C in 60 min and atmospheric pressure, BC_300/60/15—biochar produced at 300 °C in 60 min and overpressure pressure of 15 bars, BC_400/60/0—biochar produced at 300 °C in 60 min and atmospheric pressure, BC_400/60/150—biochar produced at 300 °C in 60 min and overpressure pressure of 15 bars, HTC280—biochar/hydrochar produced in hydrothermal carbonization process at 280 °C in 60 min at a pressure of up to 15 bars.

**Table 3 materials-15-00945-t003:** Low-temperature biochar properties.

Material	MY, % **	MC, % *	TS, % *	VS, % **	AC, % **	SSA, m^2^ × g^−1^	V_t_, cm^3^ × g^−1^	L, nm	pH ***	EC, mS × cm^−1^ ***
300/60/0	42.6	4.5	95.5	79.5	20.5	0.62	8.2 × 10^−4^	5.2	8.61	3.04
300/60/15	45.9	3.3	96.7	89.6	10.4	0.26	3.3 × 10^−4^	5.0	8.04	3.57
400/60/0	37.4	4.4	95.6	77.3	22.7	0.61	7.6 × 10^−4^	5.0	10.19	4.53
400/60/15	34.3	4.0	96.0	60.9	39.1	0.64	11.3 × 10^−4^	7.1	10.75	7.69
HTC280	56.4	18.4	81.6	88.1	11.9	0.38	5.6 × 10^−4^	5.9	5.59	4.71

* as-received base, ** dry base, *** measured in solution: 1 g _BC_ to 10 mL _deionized water_, after 30 min.

**Table 4 materials-15-00945-t004:** The biochar addition effect on the process residues and methane production, after 21 days.

Biochar	No.	Initial		End		Process Residues’ Properties				Mass Reduction, %	BD, %	CH_4_ Production Effect, %
		pH	EC, µS × cm^−1^	pH	EC, µS × cm^−1^	MC, %	TS, %	VS, %	AC, %			
D + FW	1	7.91	61.4	7.92	76.1	95.8	4.2	61.0	39.0	3.6	79.6	-
	2	7.85	63.6	7.92	72.7	95.6	4.4	58.9	41.1	3.7	78.2	-
	3	7.69	65.7	8.02	73.5	95.8	4.2	59.6	40.4	2.1	73.3	-
	4	7.68	65.1	7.99	73.5	95.8	4.2	60.3	39.7	2.5	71.7	-
	Mean	7.78	64.0	7.96	74.0	95.7	4.3	59.9	40.1	3.0	75.5	-
300/60/0	1	7.82	56.1	7.97	75.6	95.6	4.4	60.7	39.3	3.0	78.7	−0.2
	2	7.85	58.6	7.92	74.4	95.6	4.4	60.5	39.5	3.0	78.4	−0.7
	3	7.62	66.1	7.96	74.1	95.7	4.3	63.1	36.9	2.5	75.5	4.1
	4	7.67	66.3	7.96	74.7	95.6	4.4	61.0	39.0	2.3	72.3	−0.3
	Mean	7.74	61.8	7.95	74.7	95.7	4.3	61.3	38.7	2.7	76.2	0.7
300/60/15	1	7.85	66.1	7.93	74.1	95.6	4.4	59.5	40.5	3.1	81.9	3.8
	2	7.84	63.8	7.93	73.5	95.6	4.4	61.5	38.5	3.1	81.2	2.9
	3	7.67	66.6	8.02	75.6	95.7	4.3	59.6	40.4	2.2	72.4	−0.2
	4	7.65	65.1	8.02	74.4	95.6	4.4	62.2	37.8	2.4	72.7	0.3
	Mean	7.75	65.4	7.98	74.4	95.6	4.4	60.7	39.3	2.7	77.0	1.7
400/60/0	1	7.86	57.9	7.92	75.1	95.6	4.4	58.5	41.5	3.1	82.6	4.7
	2	7.84	65.1	7.92	75.9	95.6	4.4	59.6	40.4	3.1	81.6	3.4
	3	7.65	65.7	7.95	74.3	95.7	4.3	61.1	38.9	2.2	75.4	3.9
	4	7.64	64.5	8.01	74.5	95.5	4.5	61.0	39.0	2.3	73.9	1.9
	Mean	7.75	63.3	7.95	75.0	95.6	4.4	60.0	40.0	2.7	78.4	3.5
400/60/15	1	7.83	65.2	7.93	77.7	95.7	4.3	60.1	39.9	2.7	72.4	−8.2
	2	7.85	65.8	7.92	76.5	95.7	4.3	59.9	40.1	3.4	72.0	−0.7
	3	7.68	67.9	8.03	73.9	95.8	4.2	64.4	35.6	2.4	-	-
	4	7.67	61.6	8.00	72.5	95.6	4.4	61.8	38.2	2.3	-	-
	Mean	7.76	65.1	7.97	75.2	95.7	4.3	61.6	38.4	2.7	72.7	−4.5
HTC280	1	7.78	64.7	7.95	75.9	95.7	4.3	61.0	39.0	3.0	81.6	3.4
	2	7.82	63.2	7.93	76.8	95.6	4.4	60.3	39.7	3.9	81.5	3.3
	3	7.64	66.0	7.99	72.0	95.7	4.3	69.6	30.4	2.3	75.4	4.0
	4	7.64	67.4	8.02	68.7	95.7	4.3	61.4	38.6	3.0	-	-
	Mean	7.72	65.3	7.97	73.4	95.7	4.3	63.1	36.9	3.1	79.5	3.6

**Table 5 materials-15-00945-t005:** Kinetic of CH4 production for all experiments.

Variant	No.	k, d^−1^	$EBMP_{e},\;{\mathrm{ml}}_{{\mathrm{CH}}_{4}}\;\times \;{g_{\mathrm{VS}}}^{-1}$	r, ml_CH4_ × (g_VS_ × d)^−1^	R^2^, -
Control	1	0.265	362.13	95.89	0.997
	2	0.270	354.13	95.48	0.996
	3	0.217	348.40	75.46	0.993
	4	0.208	340.94	70.88	0.992
	Mean	0.240	351.40	84.43	0.995
300/60/0	1	0.266	357.43	95.25	0.996
	2	0.264	357.42	94.29	0.996
	3	0.205	357.16	73.31	0.995
	4	0.202	343.32	69.23	0.993
	Mean	0.234	353.83	83.02	0.995
300/60/15	1	0.281	371.93	104.62	0.997
	2	0.273	371.08	101.45	0.997
	3	0.212	342.20	72.62	0.993
	4	0.217	344.88	74.90	0.993
	Mean	0.246	357.52	88.40	0.995
400/60/0	1	0.249	377.05	93.77	0.996
	2	0.268	368.88	98.99	0.996
	3	0.200	356.88	71.20	0.994
	4	0.222	347.68	77.29	0.994
	Mean	0.235	362.62	85.31	0.995
400/60/15	1	0.250	326.62	81.75	0.995
	2	0.208	341.82	70.96	0.994
	3	-	-	-	-
	4	-	-	-	-
	Mean	0.229	334.22	76.36	0.995
HTC280	1	0.254	361.80	91.93	0.992
	2	0.238	364.77	86.82	0.992
	3	0.210	356.53	74.69	0.995
	4	-	-	-	-
	Mean	0.234	361.04	84.48	0.993

## Data Availability

All data derived during the experiments are given in the paper.

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
