# Peer review of "The Influence of Low-Temperature Food Waste Biochars on Anaerobic Digestion of Food Waste"

_materials, 2022, doi:10.3390/ma15030945_

Round 1

Reviewer 1 Report

Materials-1516479-peer-review-v1

The influence of Low-Temperature Food Waste Biochars on 2 Anaerobic Digestion of Food Waste

Review

General comments

The study presented here is clearly written and, in addition, the topic presented is of high interest. In my opinion the work is suitable for publication and I will give here some comments which, even if they are quite necessary in my opinion, they will be easily addressed by the researchers.

More specific comments

  1. In the Abstract session, it is normally not suggested to write abbreviations like FW, but write them in full
  2. The introductory part is done in a very clear way and personally I like reading it
  3. Line 93. What are “the others”?
  4. Lines 147-149. Here you could describe how these values were derived.
  5. For the same kind of proximate analyses, you could refer to some examples of literature were it is justified this characterization. For example the book chapters:

https://doi.org/10.1007/978-3-030-11599-9_6

https://doi.org/10.1007/978-3-030-11599-9_7

  1. The experimental part of this study is described clearly, however it would help if some more details are added. For example, drawings with measures etc. Photos of the equipment used with all dimensions and components. Thermocouples models, valves etc. In this regards, the char preparation from HTC should be very detailed to allow other researchers to replicate the experiments.
  2. Explain in a more clear way lines 177-179
  3. Also lines 186-188 are not very clear.
  4. Equation 1 is very elementary and maybe is not necessary
  5. When equations are elementary you can just explain them or use some references.
  6. For the equations in general, I suggest to use the Equation editor of MS Word and add some spaces between the text and the equations

Reviewer 2 Report

This study compared five types of biochars for the enhancement of FW AD. The research question is interesting, and the rationale is adequately addressed. Overall, the methods and results are clearly presented. However, the biggest issue in this study is that the results did not successfully achieve the objective of the study, enhancement of FW AD. In my opinion, there should have been some preliminary experiments to find some effective condition range where addition of biochars actually promote biogas production. Otherwise, the objective of this study only remains partly achieved. One possible alternative to this could be mass balance and energetic calculation. As the authors emphasized, this study used the same FW for both AD substrate of biochar material. So suggesting the fractions of the FW coming to the plant as substrate (for example, 90%) and biochar material (for example, 10%; that translates into the amount of biochar needed considering MY) would give some valuable idea to the readers. Also, potential energy yield based on the scenarios (i.e., 100% FW for AD case vs. 90% for FW + 10% for BC, as an example) could be useful, including the energy calculation needed to make BC.

Some specific comments are below.

- Please check for language errors throughout the manuscript.

- What temperature was the Bio-Wat plant operated at?

- Remove duplicate descriptions in part 2.1.3.

- For MY calculation, is it all dry basis (including HTC)?

- Table 2. Isn’t it just quadruplicate, not 2x duplicate? If so, simply the Table content. Otherwise, please give more explanation.

- Table 4. Define CH4 production effect. % or % point? Compared to D+FW? And please give comments why some trials have no BD and CH4 results.

- Some discussion of other studies using BC for AD would be valuable, especially that had statistically improve results. Maybe more BC (than 5%) was needed?

Round 2

Reviewer 1 Report

The authors have addressed the comments given. 

Reviewer 2 Report

The MS was adequately revised according to my comments. Please correct the following before/during the proofreading process.

- Remove ")" in L435.

- Remove "," after "Because" in L449.